# Preparation and Characterization of a Renewable Starch-g-(MA-DETA) Copolymer and Its Adjustment for Dye Removal Applications

**DOI:** 10.3390/polym15051197

**Published:** 2023-02-27

**Authors:** Lolwah Alfuhaid, Eman Al-Abbad, Shouq Alshammari, Aljawharah Alotaibi, Naved Malek, Azza Al-Ghamdi

**Affiliations:** 1Department of Chemistry, College of Science, Imam Abdulrahman Bin Faisal University, P.O. Box 1982, Dammam 31441, Saudi Arabia; 2Basic & Applied Scientific Research Center (BASRC), Renewable Energy Unit, Imam Abdulrahman Bin Faisal University, P.O. Box 1982, Dammam 31441, Saudi Arabia; 3Ionic Liquids Research Laboratory, Department of Chemistry, Sardar Vallabhbhai National Institute of Technology, Surat, Gujarat 395007, India

**Keywords:** grafting modification, starch, water treatment, celestine blue

## Abstract

Maleic anhydride-diethylenetriamine grafted on starch (st-g-(MA-DETA)) was synthesized through graft copolymerization, and the different parameters (copolymerization temperature, reaction time, concentration of initiator and monomer concentration) affecting starch graft percentage were studied to achieve the maximum grafting percentage. The maximum grafting percentage was found to be 29.17%. The starch and grafted starch copolymer were characterized using XRD, FTIR, SEM, EDS, NMR, and TGA analytical techniques to describe copolymerization. The crystallinity of starch and grafted starch was studied by XRD, confirming that grafted starch has a semicrystalline nature and indicating that the grafting reaction took place typically in the amorphous region of starch. NMR and IR spectroscopic techniques confirmed the successful synthesis of the st-g-(MA-DETA) copolymer. A TGA study revealed that grafting affects the thermal stability of starch. An SEM analysis showed the microparticles are distributed unevenly. Modified starch with the highest grafting ratio was then applied to celestine dye removal from water using different parameters. The experimental results indicated that St-g-(MA-DETA) has excellent dye removal properties in comparison to native starch.

## 1. Introduction

Environmental contamination of natural resources has been rapidly increasing in recent years, and it must be addressed as a top priority if the earth and its inhabitants are to be sustained for future generations. Water is vital for all living beings in order to survive and live. Providing safe, inexpensive, and/or clean drinking water is a huge environmental challenge all around the world.

Water is used for a variety of purposes in various industries, such as growing fruits and vegetables in agriculture, construction in civil engineering, pharmaceutical formulation and development, electricity generation in power plants, and so on. However, drinking water is an absolute necessity for living beings to survive. As a result, at present, environmentalists are concerned about water pollution, since it has a direct or indirect impact on the health of not only people but practically all species of life [1].

Many pollutants, such as heavy metals, dyes, pharmaceuticals, fertilizers, pesticides, crude oil products, phenolics, radionuclides, and other chemicals, cause contamination of surfaces and groundwater [2]. Dyes are toxic organic substances used as coloring agents and are found in significant amounts in the effluents of food, pharmaceutical, leather, paper, cosmetics, and textile industries, and they are one of the top three pollutants. The textile industry alone consumes roughly 700,000 tons of dye per year, making them major water polluters [3,4].

Dye molecules decompose in the aquatic environment, resulting in toxic materials that adsorb oxygen from the water, and this has increased to the point where it now poses a serious threat to human life and the environment [5].

There are many techniques applied to isolate dyes and other contaminants from polluted water such as adsorption, ion exchange, coagulation, precipitation, flocculation, membrane filtration, chemical reduction, and electrochemical treatment [6,7,8,9]. Adsorption is one of the most widely used techniques for removing pollutants from wastewater resources due to its simplicity, efficiency, and low cost. The adsorption capacity depends on the dye’s physical and chemical properties and chemical composition [10,11].

Several ways of developing cheaper and more effective adsorbents, including natural polymers, have recently been proposed. Polysaccharides are one class of biomolecule that have also been investigated as an effective biomolecule for the manufacturing of bio-sorbents from biomass resources, and they can be found in almost all living things, including seaweed, plants, microorganisms, and animals [12,13].

Starch is a natural polymer originating from different plant organs and is widely used in industry for various applications. Native starch contains amylopectin (70–90%) and amylose molecules (10–30%) which vary in their botanical origin [14,15]. The variation in percentage depends on the origin of the starch. Amylopectin is a more branched macromolecular component that is present in starch (70–80%), possessing between one and six additional glycosidic links. Starch use is very limited in industrial as it is partially soluble in water. Thus, native starch is often modified with organic moieties to improve its physicochemical, biodegradable, and physicomechanical properties. Chemically modified starch has a lot of industrial applications [16]. Starch is generally modified by four methods, namely genetic, chemical, physical, and enzymatic. Among them, chemical modification is preferred because modified starch has enhanced properties compared to starch molecules. Starch has three -OH groups at the C2, C3, and C6 positions, which could be graft polymerized by esterification and oxidation processes [17].

Grafted starch copolymers are becoming increasingly important because of their potential applications in industry. They have been used as hydrogels, flocculants, ion exchangers, superabsorbents, etc. Grafted starch copolymers can be made primarily by free radical initiation. The aim of this work is the preparation and characterization of a grafted copolymer by construction of organic moieties which possess active functional groups and inserting them into a starch matrix. This material is adjusted for celestine blue removal from aqueous solution, then its various parameters are examined that affect the grafting process. The starch-grafted copolymer has been systematically characterized using XRD, FT-IR, SEM, EDS, ^13^C NMR, and TGA analytical techniques. Furthermore, the prepared grafted starch will be used to adsorb celestine blue (CB) dye, and the removal efficiency of the grafted starch is compared with native starch. The effects of various conditions such as pH, contact time, adsorbent dosage, initial solution concentration, and temperature are also studied.

## 2. Materials and Methods

### 2.1. Materials

Starch (Analar BHD chemie Ltd., Poole, UK), acetic acid (East Anglia Chemicals, Fakenham, UK), potassium persulphate (≥99.0%—Sigma-Aldrich, St. Louis, MO, USA), sodium bisulfate (99.0%—Sigma-Aldrich), diethylene triamine 99% (Sigma Aldrich), maleic anhydrate (99%—Acros Organics, Somerville, NJ, USA), sodium acetate 99% (Sigma Aldrich), ethanol absolute 99% (Fisher Chemical, Waltham, MA, USA), and analytical grade chemicals and solvents were used in all the experiments.

### 2.2. Methods

#### 2.2.1. Preparation of Maleamic Acid Monomer

Maleamic acid was synthesized via the interaction between maleic anhydride (MA) and diethylenetriamine (DETA). DETA (0.05 mol) was added slowly to 20 mL of deionized (DI) water in a three-neck flask. Then, MA (0.05 mol) was slowly added to the above solution and stirred for 10 min at room temperature. After that, the mixture was heated at 95 °C for 1 h, and the obtained maleamic acid was stored in a clean glass vial for further studies.

#### 2.2.2. Preparation of St-g-(MA-DETA)

An amount of 2 g of starch was dissolved in 100 mL DI water and charged into a three-neck flask, then 3 mL acetic acid was added to the mixture and maintained at 30 °C. Then, various concentrations of redox system initiator (potassium persulphate) (0.05, 0.3, 0.6, 0.8, 1, and 2 mol/L) were added to the slurry and left for one hour to form a free radical on the starch backbone. After that, an amount of monomer (0.15, 0.2, 0.45, 0.9, and 1 mol/L) was mixed with the above solution. Then, the reaction was carried out under an inert gas environment at various temperatures (30 °C, 40 °C, 50 °C, 60 °C, 70 °C, 80 °C, and 90 °C) and reaction times (3, 6, 9, 12, and 15 h). Figure 1 illustrates a schematic description of the starch grafting process. Then, the grafted starch was precipitated in a mixture of 70 mL ethanol and 40 mL deionized water. The precipitated grafted starch was washed with DI water and dried at 45 °C for one day to get persistent weight [18,19]. Finally, the homopolymer was separated from the grafted copolymer by a Soxhlet system using methanol.

#### 2.2.3. Calculation of Grafting Percentage and Grafting Efficiency

The effects of different parameters were described by Equations (1)–(3) [20].
(1)$$\mathrm{Grafting}\;\mathrm{percentage}\;\left(\mathrm{GP}\%\right)=\frac{W_{2}-W_{1}}{W1}\times 100$$
(2)$$\mathrm{Grafting}\;\mathrm{efficiency}\;\left(\mathrm{GE}\%\right)=\frac{W_{2}-W_{1}}{W_{mon}}\times 100$$
(3)$$\mathrm{Conversion}\;\left(\%C\right)=\frac{W_{2}}{W_{mon}}\times 100$$
where *W*_1_ (g) is the mass of pure starch, *W*_2_ (g) is the mass of grafted starch, and *W_mon_* (g) is the mass of the monomer.

#### 2.2.4. Determination of Carboxyl Content

An acid-base titration was performed to estimate the number of carboxyl groups in the different samples of starch. In the first step, 0.5 g of adsorbent was submerged in 25 mL of 0.1 M HCl, then stirred with a magnetic stirrer for 30 min. In the second step, the mixture was washed with 40 mL of DI water to remove the hydrochloric acid. In the third step, the starch samples were transferred to 500 mL beakers and 30 mL of DI water was added, then the mixture was heated for 15 min with continuous stirring to ensure complete gelatinization. Finally, 0.1 mol/L of standardized NaOH and hot starch was titrated using phenolphthalein as indicator. The number of carboxyl groups per anhydroglucose unit was used to calculate the carboxyl content and the degree of substitution was estimated using Equation (4) [21,22]:(4)$$M_{eq}\;of\;acidity/100\;g\;starch=\frac{\left[\left(V_{s}-V_{b}\right)\right]\left(mL\right)\times C_{NaOH}\times 100}{Sample\;weight\;\left(dry\;basis\right)\;in\;gram}$$
where *M_eq_* is the milliequivalent, 𝑉_𝑠_ and 𝑉_𝑏_ are the volumes of sodium hydroxide consumed by the sample and the blank, respectively, and *C_NaOH_* is the concentration of *NaOH* in mg/L.
(5)$$\%\;carboxyl=[\frac{M_{eq}\;of\;acidity}{100\;g\;of\;starch}]\times 0.045$$
(6)$$DS=\frac{0.162\;\left(A\times N\right)/W}{1-\left[0.101\times \left(A\times N\right)/W\right]}$$
where *A* is the volume of NaOH in mL, *N* is the normality of sodium hydroxide, and *W* is the mass of the sample in grams.

#### 2.2.5. Preparation of Dye Solution

To make the dye stock solution (250 mg/L), 0.25 g of celestine blue dye was dissolved in deionized water. Then, a series of various concentrations was obtained by diluting the stock solutions as follows:(7)$$M_{1}V_{1}=M_{2}V_{2}$$
where *M*_1_ and *M*_2_ (mg/L) are the initial and final concentrations, respectively, and *V*_1_ and *V*_2_ (L) are the initial and final volumes, respectively.

#### 2.2.6. Dye Sorption by Grafted Starch Adsorbent

Sorption of the cationic dye by the grafted adsorbent was performed in a batch system with different parameters. A dye solution with a 15 mg/L concentration was prepared and a certain amount of adsorbent (0.05 g) was added to glass containers with 50 mL of dye solution stirred in a shaker at 25 °C, 100 rpm, and a contact time 4 h. After that, the solution was centrifuged for 20 min at 6000 rpm, then filtered. The filtrates were analyzed using a UV spectrophotometer at the maximum wavelength of λ_max_ = 644 nm. The adsorption capacity and removal percentage were estimated using the following equation:(8)$$Q_{e}=\frac{(C_{o}-C_{e})\times V}{m}$$
(9)$$RE\;\%=\frac{C_{o}-C_{e}}{C_{o}}\times 100\;$$
where $Q_{e}$ (mg/g) represents the equilibrium adsorption capacity; *C_o_* and *C_e_* (mg/L) represent the initial and equilibrium concentrations, respectively; *V* (L) represents the volume of solution; and *m* (g) represents the mass of the adsorbent.

#### 2.2.7. Regeneration Experiments

Adsorption–desorption cycles were used in the regeneration experiments on adsorbents. For each adsorption cycle, 50 mL of dye with an initial concentration of 10 ppm was applied to 0.03 g of adsorbent and shaken for 180 min. Then, the concentration of CB in the solution was estimated by using a UV/Vis spectrophotometer. The dye-loaded adsorbent was filtrated, collected, and dried at 45 °C, then dispersed in 10 mL of 0.2 M HCl and desorbed for 1 h at 1000 rpm and 25 °C, where the concentration of CB was determined by a UV/Vis spectrophotometer. The adsorbent was then recovered and reused in the subsequent adsorption cycle experiment [11].
(10)$$\%\;Desorption=\frac{\left(C_{D}V_{D}\right)100}{q_{e}m}$$
where *C_D_* (mg/L) and *V_D_* (L) are the dye concentration and volume of desorbed solution used, respectively. The mass of the adsorbent utilized for desorption is *m* (g) and the adsorption capacity of the CB loaded adsorbent is *q_e_* (mg/g).

#### 2.2.8. Characterization

The structural properties of starch and modified starch were obtained using X-ray diffraction, XRD (Shimadzu-XRD-7000, Columbia, NY, USA). Nuclear magnetic resonance spectroscopic measurements of monomers and polymeric samples were carried out on Bruker Advance-600 MHz spectrometers. The ^13^C NMR spectra of 20–50 mg purified samples dissolved in 450–500 microliters of deuterated DMSO were recorded by (Bruker spectrometer, Zurich, Switzerland). Fourier transform infrared spectroscopy (FT-IR) data were obtained on a (Shimadzu-FT-IR IRSpirit, Kyoto, Japan). The spectra were recorded in the absorption mode in the range of 4000–500 cm^−1^ with a 4 cm^−1^ resolution using KBr pellets made from a mixture of polymer (1 wt%) and KBr. pH measurements were obtained by a pH meter (Electronic Instruments Limited-7020, Columbus, OH, USA). The morphology of the prepared samples was examined using scanning electron microscopy (SEM) (VEGA3 TESCAN, Brno, Ceske Republic). Thermal gravimetry analysis was performed on a TA-Q500 (Mettler-Toledo). The adsorption efficiency of the bio-sorbent was determined using a UV/vis spectrophotometer (UH5300-HITACHI, Japan).

## 3. Results and Discussion

### 3.1. Synthesis of Grafted Starch

The synthesis of modified starch was carried out in two steps. First, the monomer was prepared via the acylation of DETA with MA to produce maleamic acid (I, Figure 2) [23]. The second step was the graft copolymerization process via three stage free radical mechanism, initiation, propagation, and termination [19]. The graft copolymerization reaction began with the initiation step, which includes the formation of free radicals through the addition of chemical initiators. The chemical initiators were added first to ensure the formation of unstable sites (macro radicals) on the starch backbone, followed by the addition of the prepared monomer (diethylenediamine maleamic acid) to initiate the graft copolymerization by building organic branches on the main chains of starch. The successive growth of the side chains by adding the organic moieties took place during the propagation step. Each addition creates a new active center of the free radical, as shown in Figure 3. Finally, the side chain growth can be terminated through combination or disproportion reactions [19].

### 3.2. Characterization Results of the Samples

#### 3.2.1. XRD Analysis

The XRD patterns of pure and chemically modified starch are presented in Figure 4. Native starch has a semi-crystalline structure with substantial reflection intensities at angles 2𝜃 of 15°, 17°, 18°, 21°, and 23° [24]. These are typical diffraction patterns of A-type starch. Short chain amylose tends to form an A-type structure, while long chain amylose tends to form a B-type structure. It can be seen that the crystallinity of starch did not change much after the grafting reaction, indicating that the reaction took place predominantly in the amorphous portions of the starch. In addition, the grafted organic moieties have active polar sites which form new secondary interaction forces between the starch chains. The intensity of the XRD diffraction peaks in the hydrophobically grafted starch was comparatively lower than those of the native starch. The relative crystallinity was measured as per article [25]; for native starch it was estimated to be 38.19%, and after modification, the grafted starch’s crystallinity was slightly lowered to 37.86%, this is attributed to the crystalline structure of starch being disrupted during graft polymerization [26]. This phenomenon was due to introducing organic moieties which led to a decrease in the ratio of hydrogen bonding between the neighboring chains. XRD analysis showed that the crystallinity was slightly affected by the grafting process relative to native starch [27]. The limited differences in the intensity between native and grafted starch might be attributed to the presence of various groups which enable secondary attractions between the monomer moieties and the starch matrix.

#### 3.2.2. Fourier Transform Infrared Spectroscopy

The FT-IR spectra reveal the existence of functional groups present in the synthesized materials. The FT-IR spectra of MA, DETA, prepared monomer MA-DETA, and the modified starch are presented in Figure 5.

The IR spectrum of MA exhibits absorptions at 1864–1860 cm^−1^, which are assigned to asymmetric stretching of the carbonyl group; an absorption at 1786–1784 cm^−1^, which is assigned to symmetric stretching of the carbonyl *v*(CO); an absorption at 1224 cm^−1^, which is assigned to the asymmetric ring stretching of =C-O-C=; and absorptions at 1064 cm^−1^ and 1051 cm^−1^, which are assigned to the symmetric ring stretching of =C-O-C=. These observations are characteristic of maleic anhydride. The band at 1663 cm^−1^ and the appearance of a band at 1562 cm^−1^ can be assigned to the stretching vibrations of the amidine group (N-C=N) and the bending vibrations of the secondary amine of DETA, respectively. MA-DETA exhibits an absorption peak at 3322 cm^−1^, which indicates the -OH groups, and the peaks at 1762 cm^−1^, 1521 cm^−1^, and 1668 cm^−1^ indicate the formation of MA-DETA.

Figure 5d shows a slightly broad peak at 3320 cm^−1^, which is due to the -OH stretching vibration of native starch [28]. The absorption band in the range of 3400–3200 cm^−1^ is attributed to the O-H link in the glycosidic ring of starch, which is able to take part in intermolecular hydrogen bonding [29]. The peaks at 2929 cm^−1^ and 1646 cm^−1^ are the bands that are related to the C-H stretching of the aliphatic ring and the existence of residual H_2_O molecules in starch, respectively, whereas the peak at 1337 cm^−1^ is assigned to -CH_2_ symmetry. The absorption band in the range of 1100–990 cm^−1^ is assigned to C-O stretching in the C-O-C and C-O-H bonds in glycosidic rings. To confirm the final structure of the monomer, FT-IR was used to verify the product that arises from the reaction between maleic anhydride and diethylenetriamine, as well as to prove the mechanism by which the interaction occurred, i.e., whether it was via a ring-opening or substitution mechanism shown in Figure 5c. Based on the observations in the following sections, an ring-opening mechanism was proposed in this study.

##### Formation of Carboxylic Acid

Due to the presence of a wavenumber shift of the carbonyl group (C=O) from 1800 and 1774 cm^−1^ (ascribed to stretching bands of symmetric and asymmetric vibrations which present in anhydrous maleic acid) to a single band at 1745.91 cm^−1^, the formation of carboxyl groups was implied.

This result is supported by the presence of a wide band ranging from 3500–3100 cm^−1^. This is might be due to the presence of stretching vibrations of acidic OH interfering with the amide or primary and secondary amines.

##### Formation of an Amide Bond

Two absorption peaks at 1569.04 cm^−1^ and 1506.28 cm^−1^, assigned to the stretching band of C=O and the bending vibration of -NH [30,31,32], confirm the presence of an amide bond. In addition, the peak at 1395 cm^−1^ is assigned as C-N stretching, while the peak at 1474.90 cm^−1^ is assigned as CH_2_ stretching.

##### Absence of Cyclic Imide

The cyclic anhydride bands (asymmetric and symmetric C=O), which are present in maleic anhydride at 1800 and 1774.4 cm^−1^, were absent in the prepared monomer (MA-DETA). Additionally, two peaks ascribed to the cyclic imide (asymmetric and symmetric C=O) at 1770 and 1700 cm^−1^ were also absent. These remarkable disappearances could be attributed to the formation of ring-opening structures under moderate conditions [31,32,33]. The double bond remains unchanged at 1623 cm^−1^.

Furthermore, the FT-IR peaks confirmed the presence of pure starch and modified starch, as well as the additional bands caused by the carboxyl, amines, carbonyl, and amide groups. Figure 3d shows a very broad band in the region of 3300–2500 cm^−1^, assigned to an overlap of the O-H peak of the carboxyl and the N-H peak of the amine. The modified starch spectrum displayed an absorption peak at 1748 cm^−1^, which belongs to the C=O stretching band of a carboxyl group [34]. Furthermore, as compared to the starch, the peak at 1430 cm^−1^ is due to the presence of a -CH_2_ in modified starch [35]. The FT-IR spectrum of modified starch shows also an increasing broadness of the peaks at 3276 cm^−1^ and 3080 cm^−1^, which is attributed to the interference of the stretching vibrations of -NH, COOH, and -CONH of the organic moieties in starch. The band at 2955 cm^−1^ is assigned to C-H stretching, the band at 1632 cm^−1^ is assigned to the C=O stretching vibration, the bands at 1536 cm^−1^ is assigned to C-N stretching (amide II), and the bands at 1103 cm^−1^ and 1038 cm^−1^ are assigned to C-O bending [36].

#### 3.2.3. Structural Characterization by NMR Spectroscopy

NMR spectra were used to elucidate the structure of the monomer and the grafted starch copolymer. Depending on the location of the peaks and the electronic environment around the atoms, the chemical structure can be confirmed using ^13^C-NMR. Figure 6 and Table 1 show the ^13^C-NMR results of the prepared monomer (DETA-MA), showing the presence of carboxyl group bands (-COOH) at 170.88 ppm, as well as the presence of an amide group (-CONH) peak at 167.74 ppm. This confirms the hypothesis that the reaction is carried out by a ring-opening mechanism. In addition, olefinic carbon atom peaks appeared at 136.61 and 131.19 ppm, which means that the double bond was not affected by the reaction between anhydrous malic acid and diethylenetriamine. A similar chemical shift has been recorded previously [37,38,39]. Aliphatic tails of the monomer moieties are found as follows [36]: first, the methylene groups (CH_2_) adjacent to the amide group appear at 40.99 ppm; second, the protons adjacent to the primary amine group (CH_2_-NH_2_) appear at 44.66 ppm; and third, the two methylene bands adjacent to the secondary amine (-CH_2_-NH-CH_2_-) have values close to 50.32 and 52.78 ppm.

The NMR spectrum of the grafted starch was studied and compared with native starch to determine the positions of newly formed or disappearing peaks to confirm that the grafting process had occurred and to help determine the appropriate reaction mechanism.

Table 2 and Figure 7 show the ^13^C-NMR results of native starch. The carbon atoms from C1 to C6 in the glycosidic ring appear clearly in the spectra. The C1 atom appears at 𝜹 = 100.4 ppm due to it being a C-anomeric atom. The peak of C4 appears at a value of 79.5 ppm, and the peaks at chemical shifts between 73 and 70 ppm are attributed to C2, C3, and C5 carbon atoms associated with hydroxyl groups. The peak at 60.8 ppm is assigned to the C6 atom which is bonded to the aliphatic hydroxyl group.

To support the determination of the grafted polymerization mechanism, the most important products are the carboxylic acid and amide groups. Figure 8 and Table 3 show the ^13^C-NMR results of grafted starch. They showed the formation of two new peaks which were not present in the pure starch spectrum. The first peak at 207 ppm belongs to the carbonyl of the carboxylic group, and the second peak at 169 ppm belongs to the carbonyl of the amide group. In addition, the peaks of the monomer moieties (CH_2_-CH_2_) appear in the range of 30–56 ppm.

The absence of double bond peaks at 136.4 and 131.19 ppm indicate that the graft polymerization process has occurred through the breaking of the olefinic bond.

The most remarkable peaks in the grafted starch spectrum are the new peaks at 57 and 65.39 ppm, which are close to the peaks of carbon C6 and C2 in the starch ring at 60.8 and 72.3 ppm. This indicates that some hydroxyl groups at position 6 could have participated in the grafting reaction [40,41,42]. No new peaks or any significant displacement or shift in the remaining carbon atoms in the glycosidic ring of grafted starch were observed. This confirms that the grafting process might occur at positions 6 and 2.

#### 3.2.4. SEM and EDS Analysis

The morphologies of starch and St-g-(MA-DETA) were analyzed by scanning electron microscopy. Figure 9 shows the microparticle size and the discerption of starch and St-g-(MA-DETA). The starch granules in Figure 9a exhibit an irregular form with smooth edges and a wide range of sizes as well as distributions, ranging from round to oval [43]. Figure 9b shows the surface morphology of the grafted starch microparticles to be uneven and their forms irregular. The morphology of the surface of pure starch shows that it is formed from oval granules. After grafting, the granule surface was covered with MA-DETA; the grafting process modified the granular structure of starch and the grafted chains surrounded and attached to the starch surface. Furthermore, the image shows the roughly spherical shape of the grafted starch, in contrast to pure starch, due to successful addition of organic monomer into the starch chains.

In addition, an EDX analysis was carried out to confirm the chemical structure and percentage of each element in native starch and grafted starch. Figure 10a illustrates the percentage composition of C (42.1%) and O (34.2%) that was obtained for native starch. Figure 10b shows the composition of C (64.7%) and O (34.9%) increased and additional new peaks related to nitrogen atoms (17.3%) are observed, which implies the efficient insertion of monomer moieties into the starch backbone.

#### 3.2.5. TGA 

TGA was performed to study the thermal behavior of the grafted starch copolymers. Figure 11 shows the thermal behavior of both starch and grafted starch with different grafting percentages. The results indicate that the grafted starch has a significant thermal property on contrast to ungrafted starch. The thermal behavior is determined through examination of T_o_ (initial decomposition temperature).

The grafted starch copolymer with a %G of 29.17% possessed a higher value of T_o_ (314 °C) than starch (248 °C, and this remarkable improvement could be attributed to the introduction of monomer moieties containing polar functional groups, such as carboxylic [44], amino, and amide groups [45], into the starch matrix. These groups are able to form inter/intra-molecular forces which give additional protection to the grafted starch chains. It was also observed that the maximum weight loss of the grafted polymer is much better (76.76%) compared with original starch (91.53%).

Figure 11a presents the TGA thermograms of starch. The first peak indicates the degradation process begins at around 80 °C, which is attributed to the presence of volatile moieties. The second peak (T_o_) is observed at 248 °C, indicating the oxidative destruction of the bio-sorbent. Correspondingly, there is a peak at about 313.88 °C, which is due to starch decomposition. Additionally, the direct degradation of starch is attributed to the breakdown of hydrogen bonding between the molecules before melting [45].

Moreover, the degradation temperature of the modified polymer, Figure 11d, was improved when compared with native starch at 314 °C. As a result, from the thermal degradation behavior, it is clear that the improvement in the degradation temperature is related to the stability of the grafted copolymer. This might be attributed to the impact of the functional groups in the monomer introduced into grafted starch matrix. The amine group is thermally stable and it takes a lot of thermal energy to start decomposing. In addition, the results revealed that the alteration produces robust bio-sorbents with a more rigid structure. Furthermore, all TGA curves of grafted starch revealed that the higher grafting percentage, the higher the thermal stability, as shown in Table 4 and Figure 11.

### 3.3. Parameters Affecting the Grafting Percentage 

The efficiency of graft copolymers for dye removal is dependent on the grafting percentage (G%). For this reason, various parameters were studied as follows:

#### 3.3.1. Effect of Reaction Time

Figure 12a presents the grafting percentage of MA-DETA onto the starch backbone with respect to reaction time. The grafting percentage for the polymerization reaction has been studied at various durations and is presented in Table 5. The results indicate that the grafting percentage increased from 3 to 9 h. Once the reaction time was extended beyond 9 h, the grafting percentage declined remarkably. The results are consistent with previous reports that as the period of polymerization increases, the grafting percentage decreases slightly before stabilizing [46]. The decrease in grafting percentage with longer reaction times is accredited to the exhaustion of MA-DETA moieties during the grafting process. In addition, this decrease might also be explained by the increase in viscosity of the reaction medium with the increase in the yield of graft copolymers, which would slow down reactant collisions and result in lower grafting ratios [47].

#### 3.3.2. Effect of Temperature on Graft Copolymerization

The maximum graft percentage was studied by ranging the temperature from 30 °C to 90 °C. Figure 12b and Table 5 summarize the results of starch grafting at various temperatures.

The ratio of grafting progressively increases and reaches a maximum at 60 °C; however, the graft percentage begins to decline when the reaction temperature is higher than 60 °C. This process can be explained by at low temperature (30 °C), the activation energy required for free radicals is low; similarly, at the optimum temperature, the activation energy is high enough to encourage the development of free radicals on starch backbone, leading to an enhanced propagation process. On the other hand, raising the temperature above 60 °C decreases the grafting percentage due to the premature termination of the growing grafted chain at a higher temperature. Additionally, at high temperatures, the formation of homopolymers increases, and this leads to an increase in the viscosity of the reaction medium and inhibits monomer movement and consequently the graft copolymerization [47,48].

#### 3.3.3. Effect of Initiator Concentration

The optimum concentration of the initiator (potassium persulphate (PPS)) was found by varying the concentration in the range of 0.05–1 mol/L. Figure 12c and Table 5 display the effect of initiator concentration on the %G with constant concentrations of starch and monomer and constant time and temperature.

Figure 12c shows the grafting percentage of starch with respect to PPS concentration. Initially, the grafting percentage starts to increase and reaches a maximum at 0.6 mol/L, then it decreases at high concentrations. This process is attributed to the increase in the active grafting sites, resulting in a higher grafting percentage.

The decrease in the grafting percentage with a high concentration of initiator is due to an excessive increase in free radicals in the starch, activating the multiple free radical coupling that leads not only to forming the homo- and co-polymer, but also to terminating the polymerization process. Therefore, based on this result, using certain initiator concentration resulted in significantly increased the grafting percentage, whereas higher initiator concentrations led to decrease the grafting percentage, this is due to the termination process which causes a shortage in the length of the chain.

#### 3.3.4. Effect of Monomer Concentration

The concentration of the monomer MA-DETA was varied from 0.15 to 1 mol/L to determine the maximum grafting efficiency. The grafting percentage increased with an increase in the concentration of MA-DETA from 0.15 to 0.45 mol/L, see Figure 12d and Table 5.

At first, the grafting percentage increased when increasing the monomer (MA-DETA) concentration and achieved a maximum at 0.45 mol/L. After that, the grafting percentage declined due to the availability of monomer molecules, which might be forming the homopolymer, and the increased viscosity of the reaction medium [49,50]

### 3.4. Determination of Acidity and the Carboxylic Acid Group

The data in Table 6 show the carboxylic acid content (COOH%) and the degree of substitution (DS%) for native starch and grafted starch (St-g-(MA-DETA)) obtained using a simple titration method. It was clear that a small number of carboxyl groups were detected in the starch due to naturally occurring hydrocarbon impurities, adsorbed organic acids, and lipids [43,51,52,53]. However, the increase in carboxyl content in modified starch indicates the successful polymerization of starch and the successful grafting of the monomer onto starch, forming hydrophobic branches around the hydrophilic starch chains.

### 3.5. pH at Point of Zero Charge

The pH at point of zero charge, pH_pzc_, of native starch and St-g-(MA-DETA) were estimated by plotting the final pH vs. the initial pH after 24 h. The value of pH_pzc_ at which the initial and final pH are equal was found to be 5.86 for native starch, while it was 6 for St-g-(MA-DETA), as shown in Figure 13. When the surface of the bio-sorbent is positively charged, the pH of the solution is lower than pH_pzc_. Alternatively, when the surface of the adsorbent is negatively charged, the pH of the solution is higher than pH_pzc_. This parameter is critical for manipulating the charge of a certain surface at a specific pH. It aids in demonstrating the most appropriate technique for removing dyes and trace metals.

### 3.6. Adsorption of Celestine Blue

The synthesized bio-sorbent (starch-g-(MA-DETA)) was used for dye (celestine blue) removal under ambient conditions. The goal of the study was to explore the effects of different parameters (pH, reaction time, bio-sorbent amount, concentration of dye and temperature) to determine the optimum parameters for dye removal using the sorption approach.

#### 3.6.1. Effect of pH

The pH of a reaction mixture is an important parameter as it affects the adsorption efficacy due to changes in the adsorbents’ surface charge and ionization level [54].

The pH range (2–10) was varied while the other parameters were fixed at a dye concentration of 15 ppm, adsorbent dosage of 0.05 g, an ambient temperature of 30 °C, and a contact time of 240 min. Figure 14a,b shows the bio-sorbent adsorption capacity and CB dye removal percentage with the increase in the pH of the reaction medium. As the adsorption capacity increases from 11.45 mg/g to 13.10 mg/g for starch and from 12.33 mg/g to 14.64 mg/g for St-g-(MA-DETA), the removal percentage also increases from 9.94% to 51.81% for starch and from 32.23% to 90.96% for St-g-(MA-DETA) in parallel with the increase in pH. A similar increase in the removal of cationic dyes has also been documented [55,56]. The St-g-(MADETA) grafted copolymer has a significant role because it is a multifunction adsorbent that contains secondary and primary amines, an amide, and carboxyl groups, which are highly affected by pH variations. At lower pH_pzc_, all these functional groups are protonated and the adsorbent surface becomes overcharged with a positive charge, producing a repulsion between cationic dye CB and the polymer surface [57]. In contrast, at higher pH_pzc_, the adsorbent surface becomes overcharged with a negative charge and this leads to the formation of strong electrostatic interactions between the polymer and the cationic dye, followed by an increase in the adsorbent capacity and removal percentage [58]. The optimum adsorption was achieved at 10 pH. For the adsorption tests, we used a pH of 7 to avoid accretion of cationic CB molecules in the hydroxide form, which is linked to higher pH values.

#### 3.6.2. Effect of Contact Time on CB Adsorption

The contact time provides useful information about the time taken to reach equilibrium. The adsorbent–adsorbate contact time was varied (10–1440 min) to determine the maximum CB dye removal percentage at 30 °C and with an initial dye concentration of 15 ppm, as presented in Figure 14c,d. It is observed that the highest adsorption capacity and removal percentage are achieved after 1440 min. The adsorption capacity increased from 13.09 mg/g to 14.05 mg/g for native starch and from 13.93 mg/g to 14.96 mg/g for grafted starch. Additionally, the removal percentage of CB dye increased from 7.47% to 54.02% and from 78.97% to 99.30% for native and grafted starch, respectively. The initial rapid increase in the adsorption capacity and removal percentage within 60 min contact time is due to the availability of vacancy sites on the bio-sorbent. After 60 min, the adsorption acceleration slowed down because of a reduction in the unoccupied sites [59]. At 120 min, equilibrium is reached, and after that, a steady state is achieved for dye removal.

#### 3.6.3. Effect of Bio-Sorbent Dosage on Biosorption of CB

The effect of St-g-(MA-DETA) dosage was investigated using 10–300 mg of sorbent to evaluate the behavior of the grafted starch toward CB dye, whereas all other conditions were constant. Figure 14e,f displays that the adsorption capacity of removal of CB by starch and St-g-(MA-DETA) decreased from 40.54 mg/g to 0.58 mg/g and from 57.44 mg/g to 0.75 mg/g, respectively. This reduction in equilibrium adsorption capacity per unit mass of adsorbent may be attributed to the greater adsorbent dose generating more available adsorption sites, resulting in unsaturated adsorption sites throughout the adsorption reaction. Additionally, this decrease could also be related to a decrease in total adsorption area and an increase in diffusion path length as a result of adsorption site overlap or accumulation [60]. The dye molecules only saturate a small fraction of the adsorbent surface and leave the remaining adsorption sites unsaturated during the adsorption reactions. On the contrary, Figure 14f shows the increase in the removal percentage of CB dye by the adsorbent from 27.38% to 63,75% for starch and from 76.59% to 99.60% for St-g-(MA-DETA) due to an increase in the sorbent surface area. These findings show that the percentage of CB removal increases with increasing sorbent concentration, reaches a maximum, and then remains nearly constant as the dosage is increased [58].

#### 3.6.4. Adsorption at a Different Initial Concentration of CB Solution

The influence of different initial CB dye concentrations on the adsorption process of the bio-sorbent is presented in Figure 14g,h. The CB dye concentraton was increased from 10 to 60 ppm and the other parameters were fixed as follows: 30 °C, 0.05 g, and 240 min. A steady increase in the adsorption capacity from 6.99 mg/g to 51.10 mg/g and from 9.02 mg/g to 55.04 mg/g was observed for native starch and St-g-(MA-DETA), respectively, Figure 14h. The bio-adsorbent surface was not fully saturated with CB molecules at high concentrations of dye solution; therefore, the adsorption capacity increased as the dye concentration increased. Furthermore, the higher the concentration of CB molecules in the solution, the greater the interaction with the active sites of the adsorbent, resulting in an improved adsorption capacity [61]. At the same time, the removal percentage of CB dye decreased from 52.17% to 31.69% for starch and from 84.37% to 61.93% for St-g-(MA-DETA) with the increasing CB concentration. The steady decline in the removal percentage with an increase in CB dye concentration is attributed to the number of available active sites on the bio-sorbent being rapidly occupied by dye molecules at low concentrations and the number of active site decreasing at higher concentrations of the dye molecule; thus, the percentage of CB removal declines [62].

#### 3.6.5. Adsorption at Different Temperatures

The adsorption process is affected by rising temperatures. Therefore, the effect of temperature was examined at varying temperatures of 25–70 °C using 0.05 g of sorbent for 240 min. Figure 15 shows the percentage of CB removed by starch and St-g-(MA-DETA) as a function of temperature. The results reveal that increasing the temperature from 25 °C to 70 °C leads to an increase in the adsorption capacity from 11.68 mg/g to 13.43 mg/g and from 12.63 mg/g to 14.53 mg/g for native starch and St-g-(MA-DETA), respectively. Additionally, the removal percentage of CB dye by starch and St-g-(MA-DETA) ranged from 11.39% to 58.23% and 53.49% to 90.70%, respectively. This might be attributed to the strong interaction between the CB molecules and the binding sites of the sorbent as the viscosity of the solution decreases as the temperature rises [63,64]. Biosorption is usually enhanced as a result of the increased surface activity and kinetic energy of the solute at higher temperatures.

### 3.7. Regeneration Performance

The measurement of how many regeneration cycles that could be reached with grafted starch is considered important. The aim of this process is to evaluate the performance of the adsorbent, the cost-effectiveness, the feasibility of future application, the economic feasibility, the environmental friendliness and the possibility of the removal of adsorbed pollutants. With 10 ppm of CB dye, a pH of 7, 0.2 M of HCl, 20 mL of HCl and 0.03 g of adsorbent, the reusabilities of native starch and St-g-(MA-DETA) were investigated, see Figure 16.

According to Figure 17, three cycles of adsorption–desorption were applied on the native starch. It showed a reduction in the dye removal percentage from 69% in the first cycle to 45% in the third cycle, and the desorption percentage was estimated to be 85.64% using Equation (10). After the third cycle, native starch was hydrolyzed in the acidic medium. The adsorption–desorption of St-g-(MA-DETA) was able to continue for thirteen cycles, and the removal percentage of CB dye decreased from 88% for the first cycle to 82% for the last cycle, with a desorption percentage of 79.81%.

These findings show that grafted starch (St-g-(MA-DETA)) has a higher stability and recyclability than native starch. It implies strong interactions between St-g-(MA-DETA) and CB molecules. In addition, St-g-(MA-DETA) bio-adsorbent could be regarded as an efficient, renewable, reusable, eco-friendly, low-cost, and stable adsorbent for dye removal applications.

## 4. Conclusions

St-g-(MA-DETA) was successfully synthesized using free radical polymerization. The prepared copolymer was characterized using FT-IR to confirm its chemical structure and it was clearly observed that organic moieties, which include carboxylic, amide and amine groups, were introduced into the main backbone of starch. A ^13^C-NMR analysis confirmed the presence of carbon atoms related to the carboxylic and amide groups in the grafted copolymer, which were absence in native starch. The surface morphology of native starch before and after the grafting process was examined by SEM analysis. The surface is smooth and contains oval granules for native starch, while it is uneven and contains roughly spherical shape particles for the grafted starch. Moreover, a TGA analysis confirmed that the thermal stability of the grafted starch was higher than native starch, at 314.83 and 248.81, respectively. The optimum parameters for obtaining the best grafting percentage were 9 h, 60 °C, 0.3 mol/L of initiator, and 0.45 mol/L of monomer. The removal of CB dye from an aqueous solution using native and grafted starch was influenced by the pH, contact time, adsorbent dosage, initial dye concentration and temperature. The regeneration experiment revealed that St-g-(MA-DETA) is regenerated using a HCl solution and could be reused thirteen times, whereas native starch could only be reused three times. These findings support the efficiency of grafting monomer moieties onto the starch backbone and the enhancement in its thermal properties. In the future, we plan to utilize the grafted starch to remove various types of pollutant, as well as to apply artificial intelligent (AI) models to estimate the performance of St-g-(MA-DETA) in removing other harmful contaminates from water.

## Figures and Tables

**Figure 1 polymers-15-01197-f001:**
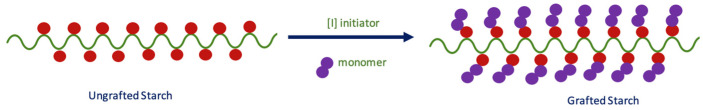
Schematic description of the St-g-(MA-DETA) synthesis.

**Figure 2 polymers-15-01197-f002:**
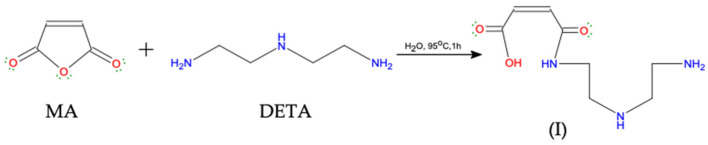
Reaction of malic anhydride (MA) with diethylenetriamine (DETA).

**Figure 3 polymers-15-01197-f003:**
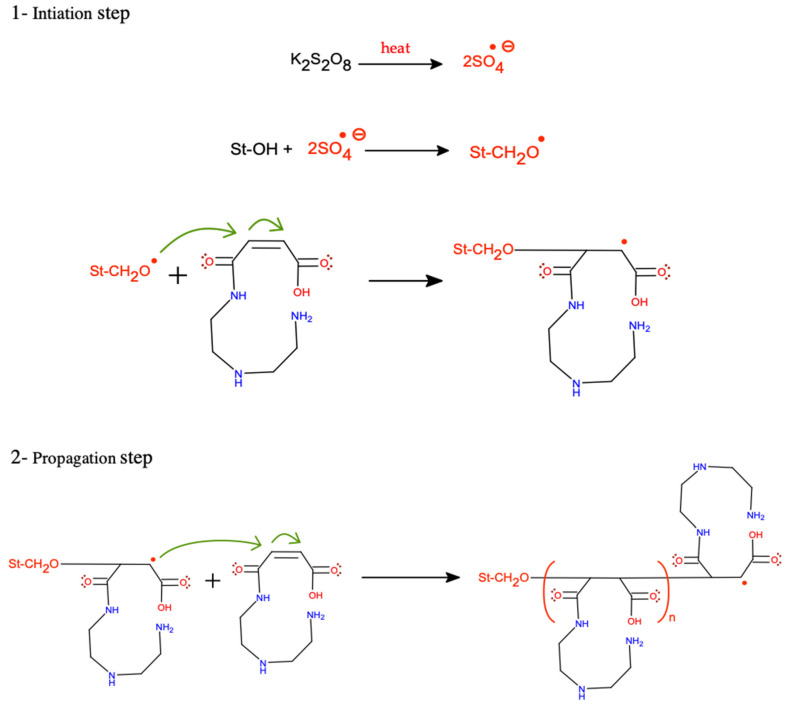
Suggested graft copolymerization mechanism of St-g-(MA-DETA), where St-OH is native starch, and St-CH_2_O**^*^** is carbon number 6 in the glucopyranose ring.

**Figure 4 polymers-15-01197-f004:**
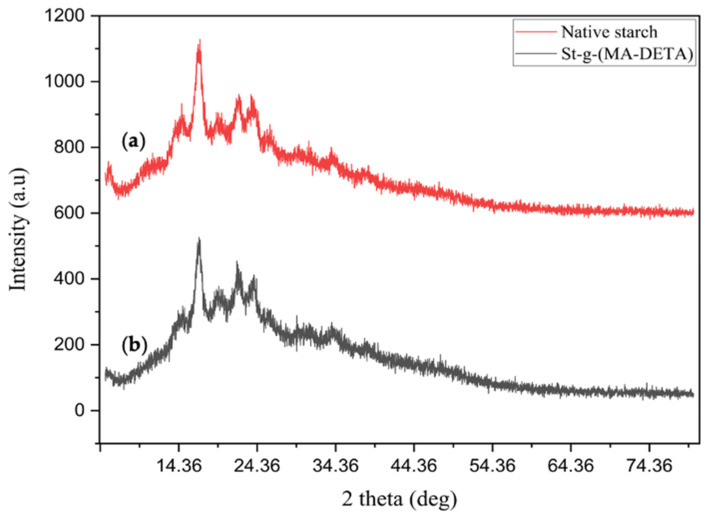
XRD diffraction of (**a**) starch and (**b**) St-g-(MA-DETA).

**Figure 5 polymers-15-01197-f005:**
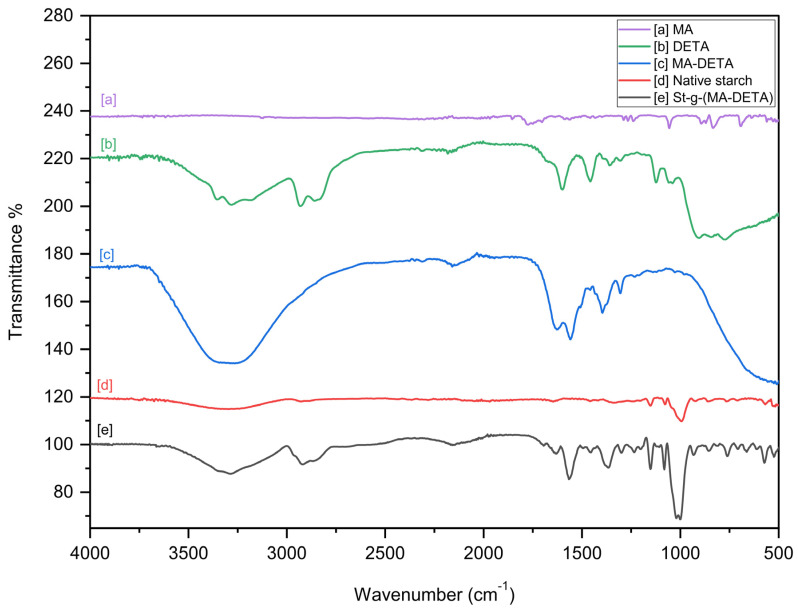
FT−IR of MA (**a**), DETA (**b**), prepared monomer (MA−DETA) (**c**), native starch (**d**) & St-g-(MA−DETA) (**e**).

**Figure 6 polymers-15-01197-f006:**
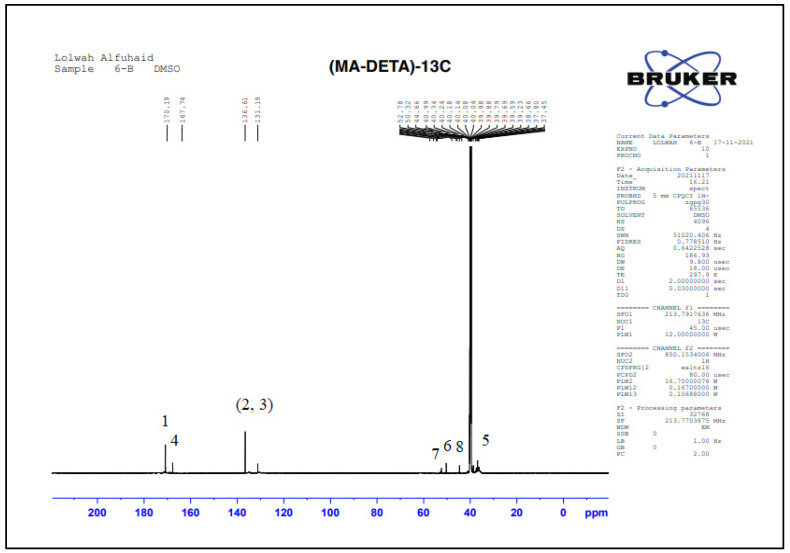
^13^C-NMR chemical shift of the prepared monomer (DETA-MA).

**Figure 7 polymers-15-01197-f007:**
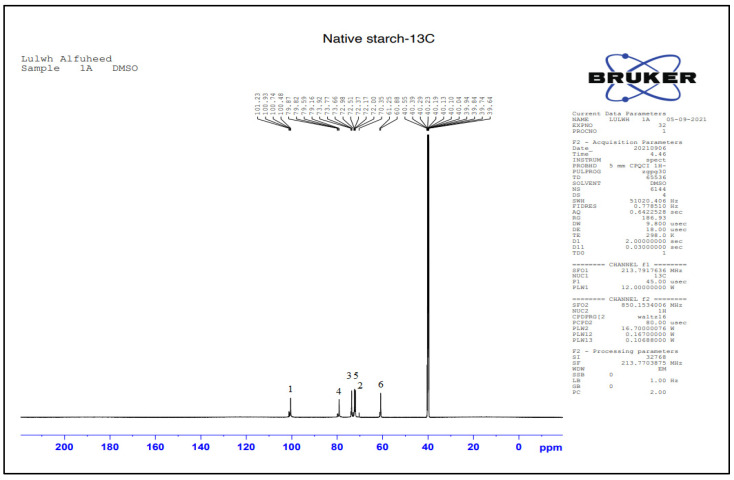
^13^C-NMR chemical shifts of native starch.

**Figure 8 polymers-15-01197-f008:**
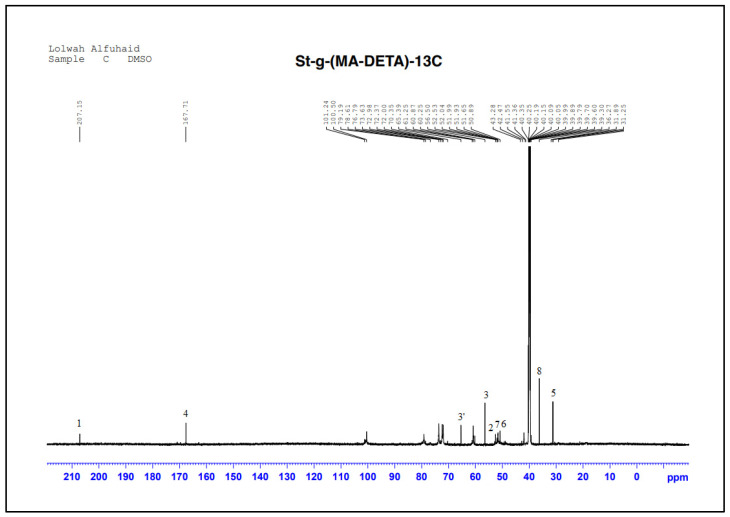
^13^C-NMR chemical shifts of the St-g-(MA-DETA).

**Figure 9 polymers-15-01197-f009:**
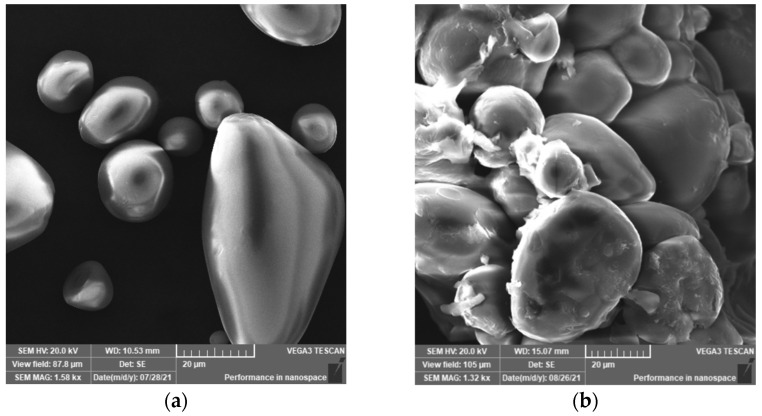
SEM images obtained for (**a**) pure starch and (**b**) St-g-(MA-DETA).

**Figure 10 polymers-15-01197-f010:**
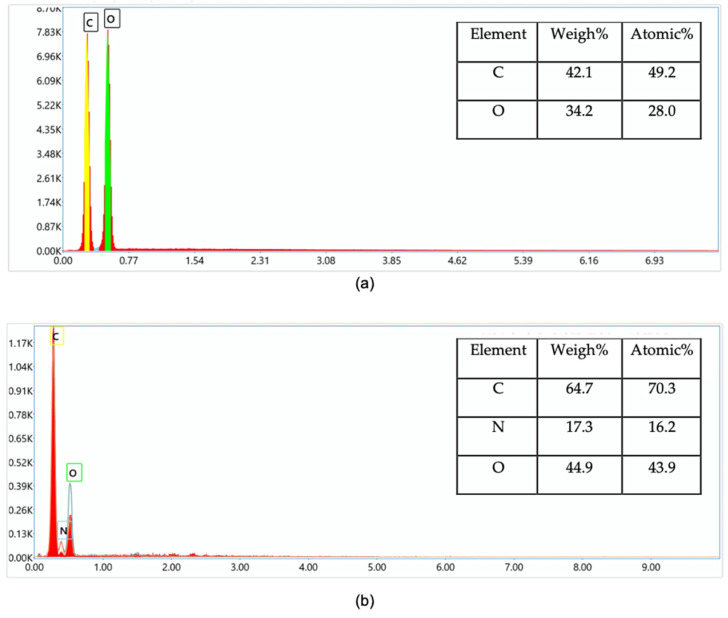
EDS spectra of (**a**) native starch and (**b**) grafted starch.

**Figure 11 polymers-15-01197-f011:**
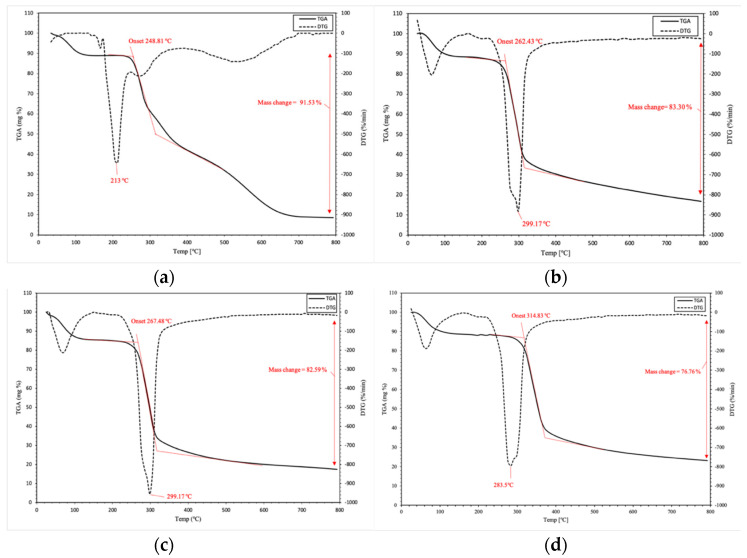
Thermogravimetric analysis (TGA) and differential thermogravimetry (DTG) of starch (**a**), St−g− (MA− DETA) (G% = 15) (**b**), (G% = 19.06) (**c**) and (G% = 27.44) (**d**).

**Figure 12 polymers-15-01197-f012:**
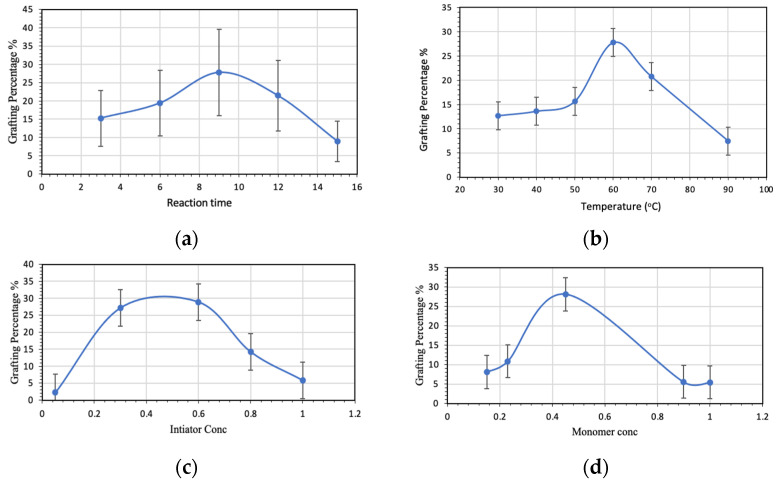
Effect of (**a**) reaction time, (**b**) temperature, (**c**) initiator concentration and (**d**) monomer concentration on the grafting percentage of starch g (MA-DETA) copolymer (starch = 2 g, [M] = 0.45 mol/L and [I] = 0.3 mol/L).

**Figure 13 polymers-15-01197-f013:**
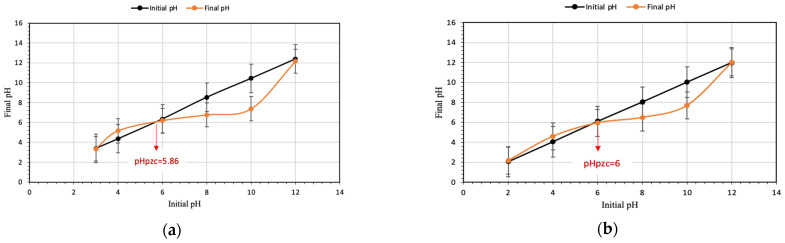
pH_pzc_ plot of native starch (**a**) and St-g-(MA-DETA) (**b**).

**Figure 14 polymers-15-01197-f014:**
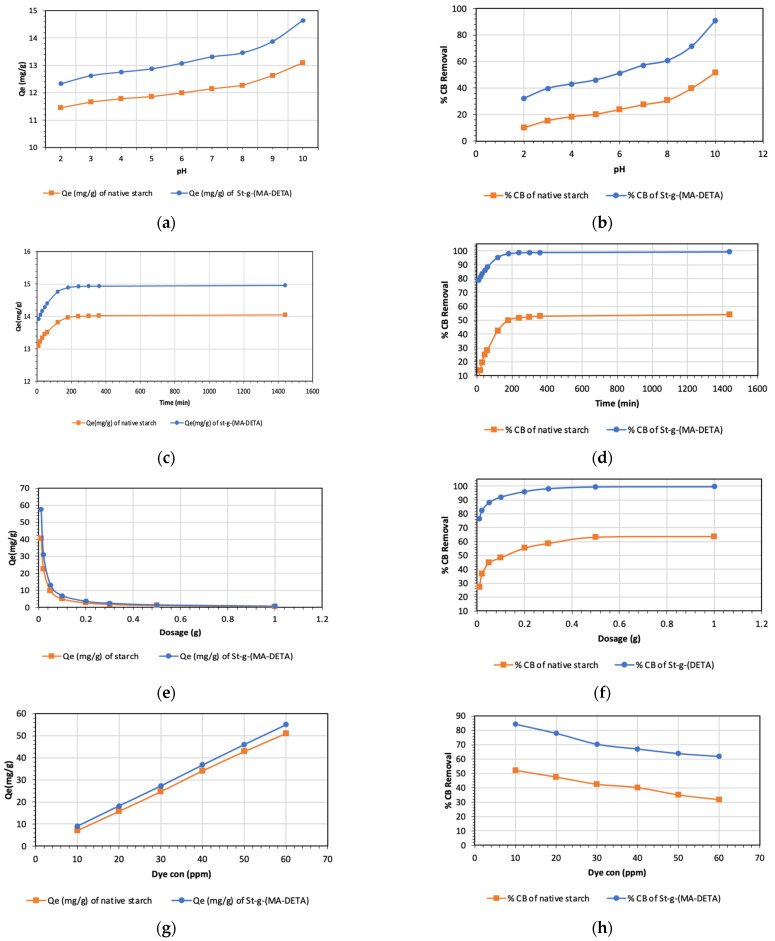
Adsorption of celestine blue from water at (**a**,**b**) different pHs (**c**,**d**) different times (**e**,**f**) different amounts of bio sorbent and (**g**,**h**) different initial dye concentrations.

**Figure 15 polymers-15-01197-f015:**
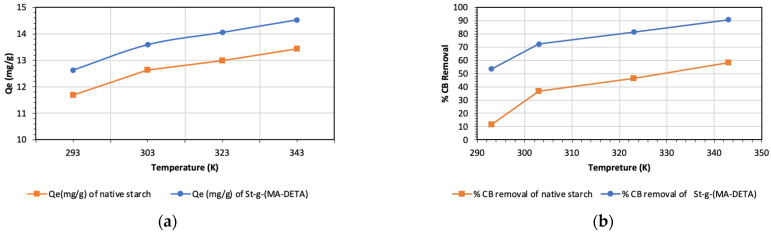
Effect of temperature on the adsorption capacity (**a**), and removal percentage (**b**) on the sorption of CB (dye concentration = 15 ppm, time = 120 min, pH = 7, adsorbent dose = 0.05 mg/L and V = 50 mL).

**Figure 16 polymers-15-01197-f016:**
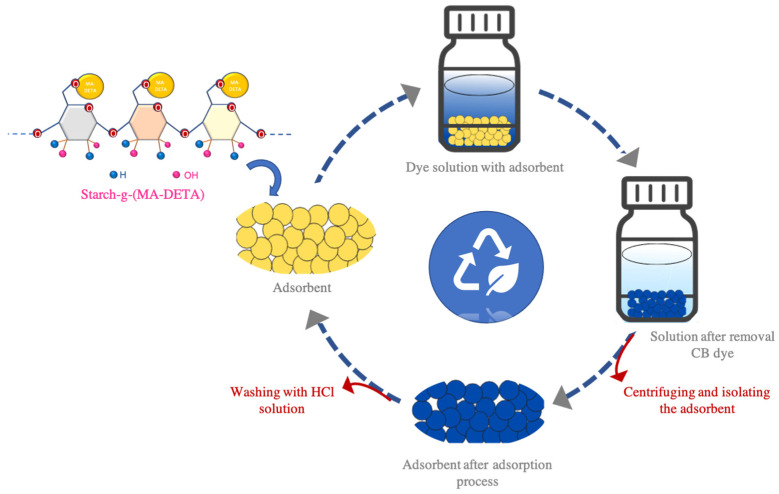
Example of the adsorption process using St-g-(MA-DETA).

**Figure 17 polymers-15-01197-f017:**
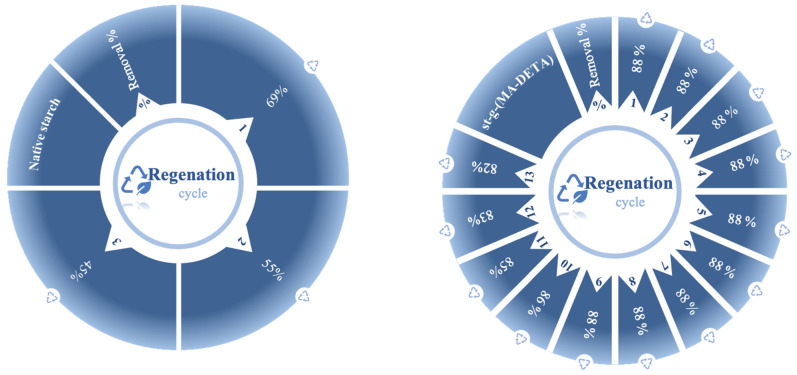
The effects of multicyclic regeneration on the rate of CB removal by different adsorbents (starch and St-g-(MA-DETA)) (10 ppm, 7pH, 0.03 g, 30 and 120 min).

**Table 1 polymers-15-01197-t001:** ^13^C-NMR chemical shifts of the prepared monomer (DETA-MA).

Suggested Structure	Carbon Atoms	Chemical Shift (ppm)
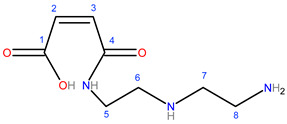	1	170.19
	2, 3	136.61–131.19
	4	167.74
	5	40.99
	6	50.32
	7	52.78
	8	44.66

**Table 2 polymers-15-01197-t002:** ^13^C-NMR chemical shifts of native starch.

Suggested Structure	Carbon Atoms	Chemical Shift (ppm)
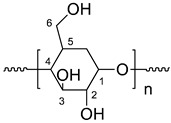	1	100
	2	72.3
	3	73.3
	4	79.8
	5	72.9
	6	60.8

**Table 3 polymers-15-01197-t003:** ^13^C-NMR chemical shifts of the St-g-(DETA-MA) copolymer.

Suggested Structure	Carbon Atoms	Chemical Shift (ppm)
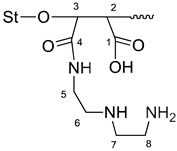	1	207.15
	2	52.53
	3	56.50 and 65.39
	4	167.71
	5	31.89
	6	50.89
	7	51.93
	8	36.27

**Table 4 polymers-15-01197-t004:** The effect of grafting percentage on the initial decomposition temperature (T_o_) value of St-g-(MA-DETA) copolymers.

Sample Name	Grafting Percentage (%)	T_o_ (°C)	Total Weight Loss (%) at 800 °C
Native starch	0	248.81	91.53
St-g-(MA-DETA)	15	262.43	83.30
	19.06	267.48	82.59
	29.17	314.83	76.76

**Table 5 polymers-15-01197-t005:** The effect of various parameters on the grafting percentage of starch g (MA-DETA) copolymer. (Starch = 2 g, [PPS] = 0.3 mol/L, T = 60 °C and time = 9 h.)

	Grafting Percentage	Grafting Efficiency	Conversion Percentage
**Time**			
3	14.95	88.97	3.91
6	19.09	98.28	4.05
9	27.44	116.63	4.34
12	21.09	102.79	4.12
15	8.63	74.24	3.69
**Temperature °C**			
30	12.34	10.98	3.82
40	13.25	11.69	3.85
50	15.32	13.28	3.92
60	27.44	21.53	4.34
70	20.42	16.96	4.09
90	7.13	6.65	3.64
**Initiator [I] mol/L**			
0.05	2.54	2.478	3.49
0.3	27.44	21.53	4.34
0.6	29.17	22.58	4.39
0.8	14.56	12.71	3.89
1	6.19	5.83	3.61
**Monomer [M] mol/L**			
0.15	7.42	6.91	68.12
0.23	10.16	9.23	76.43
0.45	27.44	21.53	93.04
0.90	4.89	4.66	8.54
1	4.74	4.52	5.76

**Table 6 polymers-15-01197-t006:** Carboxylic acid content of native starch and St-g-(MA-DETA).

Samples	Native Starch	St-g-(MA-DETA)
Grafting percentage	0	29.17
*M_eq_* of acidity/100 g of starch	2	30
Carboxyl content (%)	0.09	1.35
Degree of substitution (DS%)	0.32	5.01

## Data Availability

Not applicable.

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
