# Peer review of "Preparation and Characterization of a Renewable Starch-g-(MA-DETA) Copolymer and Its Adjustment for Dye Removal Applications"

_polymers, 2023, doi:10.3390/polym15051197_

Round 1

Reviewer 1 Report

1. Chemical formula such as CH3COOH shall be written according to the standard.

2. Whether there are spaces between numbers and units shall be unified.

3. The DET letter in Figure 2 is blocked

4. The pictures in Figure 9 and Figure 10 appear repeatedly.

5. Many data in the chart do not have standard deviation and significant difference analysis, present the original data and need to be modified.

6. Figures 12 and 13 have no error line and do not conform to the specification

7. The problems of parallel and significant difference analysis of samples appear repeatedly in various charts throughout the article.

Author Response

Response to Reviewer 1 Comments

Point 1:Chemical formula such as CH3COOH shall be written according to the standard.

Response 1: Thank you for your comment, it is corrected in the manuscript (page 3 – line 111) with a green highlight.

Point 2: Whether there are spaces between numbers and units shall be unified.

Response 2: Thank you for your comment: it may be due to a technical issue, but the manuscript is reviewed to avoid any typing mistakes.

Point 3: The DET letter in Figure 2 is blocked.

Response 3: Thank you for your comment, it is corrected (page 7 – line 231 ).

Point 4:The pictures in Figure 9 and Figure 10 appear repeatedly.

Response 4: Thank you for your comment: the EDX values are extracted from the SEM figure. The repeated pictures are omitted. (page 15 – line 465 )

Point 5:Many data in the chart do not have standard deviation and significant difference analysis, present the original data, and need to be modified.

Response 5: Thanks for your comment, standard deviations are added in Figures 12 & 13 (pages 18 & 21)

Point 6: Figures 12 and 13 have no error line and do not conform to the specification

Response 6: Yes, according to fruitful suggestion, error bars are added and highlighted in the manuscript (page 21 -line 262)

Point 7: The problems of parallel and significant difference analysis of samples appear repeatedly in various charts throughout the article.

Response 7: Thank you for your comment, could you please mention the repeated part in the manuscript?

*Note: we revised the introduction section and omit the repeated information.

Reviewer 2 Report

Dear Authors 

The presented work in the manuscript is interesting from the point of development of biobased adsorbent for wastewater treatment. To improve the quality of the work and deliver more clear information, the following comments and suggestions should be considered during the revision of the manuscript.

Title: The title should indicate what type of contaminant has been treated in the wastewater.

Abstract: The abbreviations such as MA and DETA should be fully defined for the first time their appearance in the text.

Introduction:

The aim of the work mentioned at the end of the introduction dismissed the application of the developed grafted starch. Please mention the application. 

2.2.1. Preparation of Maleamic acid

Should be modified to "2.2.1. Preparation of Maleamic acid monomer."

The chemical reaction of the monomer preparation and its structure should be provided. 

2.2.2. Preparation of St-g-(MA-DETA).

The components of the "redox system initiator" should be mentioned. 

2.2.3. Calculation of grafting percentage and grafting efficiency

Is Equation 3 present the amount of monomer converted to polymer (grafted polymer and homopolymer)??. Please declare or remove.

2.2.6. Dye sorption by the grafted starch adsorbent

The authors mentioned, " After that, the solutions were centrifuged for 20 min at 60rpm then filtrates".  I think that the centrifuge speed is 6000 rpm, not 60 rpm. Please correct.

3.1. Synthesis of grafted starch

The authors should mention in detail the method of the St-g-(MA-DETA) separation from homopolymer.

3.3. Determination of acidity and the carboxylic acid group

The acidity of the St-g-MA-DETA with different grafting percent should be provided. Also, the data presented of the st-g-(MA-DETA) in Table 5 should be labeled with the grafting percentage.

3.6. Adsorption of Celestine blue

The adsorption capacity values should be revised. The calculation of the adsorption capacity at different pH studies is incorrect.  Please revised. 

Also, the adsorption capacity in the study of the adsorbent dose effect is incorrect since the values of starch and grafted starch are almost identical where the removal percentage is clearly in favor of the grafted starch sample ????. Please revise and correct. 

The same we can tell about the adsorption capacity of the dye concentration studies. 

3.7. Regeneration Performance

The data presented in Figure 16 are completely different from that cited in the text. Please revise and correct. 

4. Conclusions

should provide the obtained results in numbers at the optimum conditions. In its current form, it looks abstract. Please rewrite.

In conclusion, a Major deep revision is needed before reconsidering the manuscript for publication. 

Author Response

Response to Reviewer 2 Comments

  1. Point 1: Title: The title should indicate what type of contaminant has been treated in the wastewater.

Response 1: Thank you for your suggestion, the title is corrected (page 1-line 1-4) with a pink highlight to be,

Preparation and characterization of renewable starch-g-(MA-DETA) copolymer and its adjustment for dye removal application

  1. Point 2: Abstract: The abbreviations such as MA and DETA should be fully defined for the first time as their appearance in the text.

Response 2: Thank you for your comment, we defined the abbreviation with a pink highlight (page 1 line 15).

  1. Point 3: Introduction: The aim of the work mentioned at the end of the introduction dismissed the application of the developed grafted starch. Please mention the application. Response 3: Thank you for your comment, we add the application of grafted starch with a pink highlight (page 3 - line 88).

  1. Point 4: “2.1. Preparation of Maleamic acid” should be modified to "2.2.1. Preparation of Maleamic acid monomer."

Response 4: Thank you for your suggestion, we modify the title with a pink highlight (page 3 – line 101).

  1. Point 5: The chemical reaction of the monomer preparation and its structure should be provided.

Response 5: Thank you for your comment, we mention the chemical reaction in “Figure 2. The reaction of Malic anhydride (MA) with diethylenetriamine (DETA)” I highlighted the figure with pink color (page 7- line 231 ).

  1. Point 6: 2.2. Preparation of St-g-(MA-DETA). The components of the "redox system initiator" should be mentioned.

Response 6: Thank you for your comment, we mention the redox system initiator which is “potassium persulfate” with a pink highlight (page 4 – line 126).

  1. Point 7:2.3. Calculation of grafting percentage and grafting efficiency Equation 3 presents the amount of monomer converted to polymer (grafted polymer and homopolymer)??. Please declare or remove.

Response 7: Thank you for your comment, it means the amount of grafted copolymer and we rewrite the equation in the manuscript with a pink highlight (page 4 - line 147).

The conversion percentage of the polymerization process was estimated by polymer weight yield percentage. After purification, the copolymer is dried to constant weight. The dried polymer weight was divided by the starting monomer weight and multiplied by 100%.

Ref: https://doi.org/10.3390/polym13183084

  1. Point 8: 2.6. Dye sorption by the grafted starch adsorbent. The authors mentioned, " After that, the solutions were centrifuged for 20 min at 60rpm then filtrates". I think that the centrifuge speed is 6000 rpm, not 60 rpm. Please correct.

Response 8: Thank you for your comment, yes it is an incorrect typing error and we corrected it in the manuscript with a pink highlight (page 5 - line 171).

  1. Point 9:1. Synthesis of grafted starch. The authors should mention in detail the method of the St-g- (MA-DETA) separation from homopolymer.

Response 9: Thank you for your suggestion, the method of homopolymer separation is added in the methodology section “2.2.2. preparation of St-g-(MA-DETA)” (page 4 -line 120) with a pink highlight.

  1. Point 10: Determination of acidity and the carboxylic acid group. The acidity of the St-g-MA-DETA with different grafting percent should be provided. Also, the data presented of the st-g-(MA- DETA) in Table 5 should be labeled with the grafting percentage.

Response 10: Thank you for your comment, this study is depending on the efficiency of the highest percentage of grafts used for dye removal application. For this reason, the acidity was only studied for the highest (%G) 29.17% of the grafted copolymer. The highest %G and % acidity are mentioned in table 6

  1. Point 11: The adsorption capacity values should be revised. The calculation of the adsorption capacity at different pH studies is incorrect. Please revised.

Response 11: Thank you for your comment, we reviewed the adsorption capacity and we didn’t find any mistakes. Could you please give us more clarification about your point of view? The adsorption capacity of the grafted copolymer at different pH is calculated according to the following equation:

Also, the adsorption capacity in the study of the adsorbent dose effect is incorrect since the values of starch and grafted starch are almost identical where the removal percentage is clearly in favor of the grafted starch sample ????. Please revise and correct.

The same we can tell about the adsorption capacity of the dye concentration studies.

Response 11: Thank you for your comments, the adsorption capacity in the study of the adsorbent was corrected with a pink highlight (page 22-line 665) and Figure 14e.

  1. Point 12: 3.7. Regeneration Performance. The data presented in Figure 16 are completely different from that cited in the text. Please revise and correct.

Response 12: Thank you for your comment, yes it is different because the number in the text represents the desorption percentage (Desorption: is the physical process where a previously adsorbed substance is released from a surface. ), while the number in the figure represented the removal percentage for each cycle (Removal Efficiency: it reflects the balance between measured values of input and output, We clarified this point in the manuscript with a pink highlight (page 26).

  1. Point 13: 4. Conclusions . should provide the obtained results in numbers at the optimum conditions. In its current form, it looks abstract. Please rewrite.

Response 13: Thank you for your comment, we add more details to the conclusion with pink highlight (page 27– line 765).

In conclusion, a Major deep revision is needed before reconsidering the manuscript for publication. Thank you for your suggestion I will take that into consideration.

Reviewer 3 Report

The manuscript by Alfuhaid et al. describes a starch-based maleamic acid grafted polymer system developed by considering various synthetic factors like copolymerization temperature, reaction time, the concentration of initiator, and monomer concentration affecting starch graft percentage which is analysed by various techniques. Finally, the modified starch with the highest grafting ratio was then applied to remove celestine 26 dye from water using different parameters. The materials are well-characterised and studied. According to me, the manuscript can be considered to be published after addressing the following comments,

  1. The introduction section looks to me very broad and not focused. The authors start with water purification, then goes to starch and grafted starch. But the real objective behind grafting starch polymer with maleamic acid is not addressed properly. Also, there is not much description of the real prototype presented here. Is it a membrane system for dye removal? If so, the effect of grafting on other factors such as hydrophilicity and mechanical properties needs to be addressed at least with some literature examples and should be reflected in the results. 
  2. What is the molecular weight of starch used in this approach? The authors need to address changes in molecular weight too with different grafting parameters.
  3. I believe there is a chance of intrachain cyclic amide formation between the free amine and carboxylic acid in the grafted polymer. How do the authors address the chance of this reaction? This needs to be clear 
  4. The XRD pattern shows an almost similar trail in grafted and nongrafted starch. The authors mention that ‘’ The relative Crystallinity was measured as per the article published by [29], for native starch it was estimated to be 38.19%, and after modification, the grafted starches were significantly lowered to different degrees attributed to disrupting the crystalline structure of starch during in situ polymerization [30].’’ Is this crystallinity value quantitatively determined from XRD data of different grafting percentages of Sta-MA-graft here based on this literature procedure in Reference 29 and 30? 
  5. A contact angle measurement for determining the change in hydrophobicity of starch membrane is suggested which is very significant concerning the objective of the material presented here. 
  6. What is the orange and blue representation in Figure 12? Please explain. 
  7. Either Figure 13 or Table 6 needs to be avoided in the manuscript as it represents the same data. 
  8. The authors need to explain the prototype of the dye removal system. Is it a membrane used and how the dye removal was conducted? 
  9. A plausible mechanism behind the recyclability of the grafted starch system here needs to be explained which will be helpful for the readers.  

Author Response

Response to Reviewer 3 Comments

Point 1: The introduction section looks to me very broad and not focused. The authors start with water purification, then goes to starch and grafted starch. But the real objective behind grafting starch polymer with maleamic acid is not addressed properly. Also, there is not much description of the real prototype presented here. Is it a membrane system for dye removal? If so, the effect of grafting on other factors such as hydrophilicity and mechanical properties needs to be addressed at least with some literature examples and should be reflected in the results.

Response 1: Thank you for your comment, we revised the introduction section to be more focused.

The real objective behind the grafting of starch with maleamic acid is clarified on (page 3 – line 84) with a blue highlight.

It is an adsorption system by using native and grafted starch as a powder. So, we did not measure the mechanical properties and did not mention that in the text.

Point 2: What is the molecular weight of starch used in this approach? The authors need to address changes in molecular weight too with different grafting parameters.

Response 2: Thank you for your comment, as you know polymeric materials have a variety of molecular weights with different chain lengths. So, our study as well as the majority of such studies confirm the grafted copolymerization by using %G calculation. In addition, we confirmed the grafting process by spectroscopic characterization such as FT-IR and NMR. Spectroscopic analysis approved the absence of some groups and appear others that were not available in starch before grafting copolymerization.

Point 3: I believe there is a chance of intrachain cyclic amide formation between the free amine and carboxylic acid in the grafted polymer. How do the authors address the chance of this reaction? This needs to be clear.

Response 3: Thank you for your comment, based on the NMR spectrum, the presence of peaks that belong to carboxylic acid and amide revealed no cyclization reaction. In addition, the closed ring structure can be formed under the influence of limited conditions such as at high temperatures or in the presence of Thionyl chloride SOCl2. Often the ring-closing reaction is not complete, and a portion of the open ring structure remains clearly measurable [Ref 1, 2].

              Ref.1 N. M. Mahmoodi, F. Najafi, and A. Neshat, “Poly (amidoamine-co-acrylic acid) copolymer:        Synthesis, characterization and dye removal ability,” Ind Crops Prod, vol. 42, no. 1, pp. 119–125, 2013, doi: 10.1016/j.indcrop.2012.05.025.

              Ref. 2 Z. Jin et al., “Modification of Poly(maleic anhydride)-Based Polymers with H2N-R Nucleophiles: Addition or Substitution Reaction?,” Bioconjug Chem, vol. 30, no. 3, pp. 871–880, Mar. 2019, doi: 10.1021/acs.bioconjchem.9b00008.

Point 4: The XRD pattern shows an almost similar trail in grafted and nongrafted starch. The authors mention that ‘’ The relative Crystallinity was measured as per the article published by [29], for native starch it was estimated to be 38.19%, and after modification, the grafted starches were significantly lowered to different degrees attributed to disrupting the crystalline structure of starch during in situ polymerization [30].’’ Is this crystallinity value quantitatively determined from XRD data of different grafting percentages of Sta-MA-graft here based on this literature procedure in References 29 and 30?

Response 4: Thank you for your comment, yes the crystallinity percentage was determined from the XRD data for the grafted starch which has the highest grafting percentage and it was 37.86%. I added this value in the manuscript (page 8 – line 250) with a blue highlight.

Point 5: A contact angle measurement for determining the change in hydrophobicity of starch membrane is suggested which is very significant concerning the objective of the material presented here.

Response 5: Thank you for your comment. Unfortunately, we did not measure the contact angle because the instrument didn't available in our lab. In addition, we didn't synthesize membrane, it is an adsorption process.

Point 6: What is the orange and blue representation in Figure 12? Please explain.

Response 6: Thanks for your comment, the zero charge analysis was reformatted, where the curve is the final pH and the straight line is the initial pH.

Plot the initial pH as (X) and the final pH as (Y), then draws a vertical line to cross the (X) axis at the point where they are equal. It is clarified on (page 21- Figure 13).

Point 7: Either Figure 13 or Table 6 needs to be avoided in the manuscript as it represents the same data. (Note: we correct the  figures numbers 13 to 12 and table 6 to 5)

 Response 7: Thank you for your comment, Figure 12 represents the relationship between various parameters vs grafting percentage to achieve the highest % of graft (29.17 %), while table 5 includes important information about the grafting efficiency and conversion percentage for each parameter at different %G. For this reason, we cannot avoid Figure.12 and table 5.

Point 8: The authors need to explain the prototype of the dye removal system. Is it a membrane used and how the dye removal was conducted?

Response 8: Thank you for your comment. No, it is an adsorption process via powder adsorbent. The following picture shows the adsorption system.

Shaking for 2h at 30 oC

(Note: The volume of the dye solution in the pictures is a part of the essential dye solution).

Point 9: A plausible mechanism behind the recyclability of the grafted starch system here needs to be explained which will be helpful for the readers.

Response 9: Thank you for your comment recycling mechanism of the grafted starch is clarified in the manuscript (Figure 16) (page 26– line 738 ).

Round 2

Reviewer 1 Report

The authors have revised the manuscript according to the review suggestions, and I suggest accepting the manuscript.

Author Response

Thank you so much, we appreciate your acceptance.

Reviewer 2 Report

Dear Authors

The comments have been answered satisfactorily by the authors.

However, the following point(s) need to be revised:

1- The adsorption capacity values should be revised. The calculation of the adsorption capacity at different pH studies is incorrect.

According to section 2.2.6 "A dye solution with a 15mg/L concentration was prepared and a certain amount of adsorbent (0.05g) was added to glass containers with 50mL of dye solution". The adsorption capacity is calculated according to yo equation 8,

Co ppm

pH

St

St-g-(MA-DETA)

R (%) / Ce ppm

Ads. capacity

R (%) / Ce ppm

Ads. capacity

15

2.0

9.94 / 13.509

1.491

32.23 / 10.166

4.8345

15

10

54.82 / 6.777

8.223

90.96 / 1.356

13.644

Accordingly, the mentioned values of the adsorption capacity by the authors need to be revised and corrected.

Author Response

Response: Thank you for your comment, the following file explains the adsorption capacity calculations.

Sincerely

Best Regards

Azza Al-Ghamdi

Round 3

Reviewer 2 Report

Dear Authors

Thanks for your explanations of the calculation methods you used.

According to what you mentioned about the calculation of the Ce, I can confirm that you used the wrong equation and I am sorry to say that.

To use the calibration curve, the following equation should be used to determine the concentration of the dye in the solution after the completion of the adsorption process (Ce);

Concentration (Ce) = Absorbance x ( 1 / Slope)   [1]

1 / Slope = 1 / 0.0843 = 11.862 

according to this value (constant), equation 1 should be:

 Concentration (Ce) = Absorbance x 11.862 

Using the obtained absorbance values you mentioned, the Ce and adsorption capacity should be as follows taking into account the adsorption capacity equation will be:

Qe = (Co - Ce) x ( V / m)    [2]

where V = 0.05 L and m = 0.05 g

So, equation 2 will be as follow:

Qe = (Co - Ce) x 1   [3]

Co ppm

pH

St

St-g-(MA-DETA)

Ce ppm

Ads. capacity

Ce ppm

Ads. capacity

15

2.0

3.546

11.45

2.67

12.33

15

10

2.135

12.86

0.356

14.644

Accordingly, I strongly recommend the revision of all the calculations in the whole manuscript regarding the adsorption capacity values. 

Author Response

Thank you for your comment. The revised manuscript was corrected and all adsorption values in the text (green highlight) and figures were updated based on your given calculation.

Round 4

Reviewer 2 Report

Dear Authors

The corrected version has been considered the raised comments and sugessions.

Accordingl;y, I can recommend the publication of the current version of the revised manuscript.

Greetings